# Enabling Better Physical Activity and Screen Time Behaviours for Adolescents from Middle Eastern Backgrounds: Semi-Structured Interviews with Parents

**DOI:** 10.3390/ijerph182312787

**Published:** 2021-12-03

**Authors:** Nematullah Hayba, Yumeng Shi, Margaret Allman-Farinelli

**Affiliations:** Discipline of Nutrition and Dietetics, Charles Perkins Centre, University of Sydney, Sydney 2006, Australia; yshi7693@uni.sydney.edu.au (Y.S.); margaret.allman-farinelli@sydney.edu.au (M.A.-F.)

**Keywords:** adolescents, parents, interviews, physical activity, screen time, obesity, overweight, prevention interventions, ethnic minority

## Abstract

The unrelenting obesity pandemic in Middle Eastern (ME) adolescents living in Australia warrants culturally responsive and locally engineered interventions. Given the influence of parents on the lifestyle behaviours of adolescents, this qualitative study aimed to capture the opinions of ME parents on the barriers and enablers to sufficient physical activity and limiting screen time behaviours in adolescents. Semi-structured interviews were conducted with 26 ME parents (female) aged 35–59 years old, most of whom resided in lower socioeconomic areas (*n* = 19). A reflexive thematic analysis using the Theoretical Domains Framework and the Capability, Opportunity, Motivation-Behaviour model was performed for coding. Parents voiced confidence in their knowledge of the importance of physical activity and limiting screen time but were less optimistic in their ability to enable change in behaviours, especially for older adolescents without outside support. Despite adolescents having the necessary skills to engage in a wide array of sports, the parents admitted deep fears regarding the safety of the social environment and restricted their children’s independent mobility. Gender differences were noted, with parents reporting older girls expressing disinterest in sports and having limited physical opportunities to participate in sports at school. It may be that a community-based participatory framework is needed to improve physical activity opportunities and to address specific physical, social, and cultural barriers.

## 1. Introduction

The continuing burden of obesity [1] is not shared but is concentrated in populations from ethnic and racial minority backgrounds [2] and lower education and income levels [3]. This is reflected internationally, where youth such as non-Hispanic black (22%) and Hispanic (26%) youth in the United States (US) exhibit a higher obesity prevalence compared to those from non-Hispanic white backgrounds (14%) [4]. Similarly, an analysis of the UK Millennium Cohort Study revealed that Black Caribbean males were three more times likely to be overweight and to have a significantly higher BMI than white adolescents. Black African females were also found to have a significantly higher BMI than white females [5]. A series of population data has shown ethnicity to be an independent mediator of unhealthy weight in children [6]. Despite these trends, obesity prevention research is lacking in recruiting adolescents from ethnic minorities [7] and/or in analysing and reporting findings by ethnic group. Furthermore, the majority of interventions have failed to produce sustained weight management [8].

In Australia, the most recent extensive survey conducted in the most populated state New South Wales (NSW) revealed that more than one quarter (27.4%) of adolescents aged 13 to 16 years old from secondary schools were categorised as being overweight or obese [9]. More importantly, the prevalence was exacerbated in specific ethnic groups, with almost half of the adolescents from Middle Eastern (ME) cultural backgrounds (41.1%) having a body mass index in the overweight or obese category compared to those from English speaking backgrounds (26.1%) [9]. The survey also indicated that adolescents from ME backgrounds were less likely to meet key physical activity and screen time targets, resulting in less adolescents reaching the Healthy Fitness Zone (39% vs. 61%), an increased lack of awareness of recommended daily screen time limits (36% vs. 58%), and more frequent eating in front of the television (37% vs. 19%) compared to those from English-speaking backgrounds [9]. In the US, almost 92% of Hispanic/Latino youth were insufficiently active and were well below the national recommendations and spent more than two hours per day on screen time [10].

Lifestyle interventions targeting adolescents are essential to favourably alter the trajectory towards adult obesity and to prevent chronic disease. Physical activity behaviours have been shown to track into adulthood [11], with lower levels linked with a predisposition for overweight and obesity as well as for other co-morbidities [12] and risk factors such as hypertension [13]. Furthermore, there is preliminary evidence supporting exercise-focused interventions for adolescents [8].

Adolescence provides a key window of change for the mitigation of risk factors and enablement by positive lifestyle interventions [14] to overturn the current early onset of chronic disease and adult obesity [12]. Although adolescents experience growing independence and autonomy from the family unit, parents are important stakeholders in shaping behaviour and in determining the home environment [15] and physical activity behaviours [16,17,18,19]. For example, population data have shown that parenting practices such as having no screen time rules were associated with overweight and obesity in adolescents [20]. Hence, in the pursuit of a culturally nuanced and effective lifestyle intervention, parents are important collaborators to inform the current adoption of physical activity and screen time behaviours in adolescents. Currently little is known about how ME parents see their role and ownership of programs to improve physical activity levels, limit screen time, and modify other lifestyle factors.

The Capability Opportunity and Motivation Behaviour Model (COM-B) [21] and the Theoretical Domain Framework (TDF) [22] were selected as the overarching theories of change to guide this study. The COM-B Model postulates that the performance of a behaviour is the result of the interaction between the person’s capability (psychological and physical), opportunity (physical and social), and motivation (reflective and automatic) [21]. The TDF puts forward 14 sub constructs that are used in conjunction, further illuminating the driving factors for each COM-B domain [22].

Hence, this exploratory study aimed to capture the perceptions and practices of the Middle Eastern parents of adolescents that may enable or prove to be barriers to their children’s physical activity behaviours and screen time behaviours. In particular, we sought to determine the current level of knowledge and concern about amounts of physical activity and sedentary behaviours as well as the physical environment and social supports that might prohibit or enable physical activity for this group. The parents’ opinions on possible intervention programs were probed.

## 2. Materials and Methods

### 2.1. Study Design

This exploratory study with parents of adolescents from ME backgrounds aimed to capture information on physical activity and screen time behaviours using semi-structured interviews. Information on food behaviours and parent perceptions of the obesity pandemic were also captured but reported elsewhere [23]. This qualitative study design was selected as it enabled in-depth opinions to be shared by participants in addition to its established high validity and reliability [24,25]. The Consolidated Criteria for Reporting Qualitative Research (COREQ) [26] was used to report on this study. This study was approved by the University of Sydney Human Research Ethics Committee (2020/708).

Two complementary behavioural frameworks were selected to guide the interview questions and subsequent coding and analysis. The COM-B model predicates that three factors and their interaction are needed for a behaviour to be performed—capability, opportunity, and motivation [21]. These are then further classified into six domains: psychological capability, physical capability, physical opportunity, social opportunity, automatic motivation, and reflective motivation. This model allows researchers to identify which components need to be altered in order to elicit the targeted behaviour [27]. Greater insight is afforded with the use of the TDF [22,28,29], which presents 14 domains that readily correspond to the domains of the COM-B model. Table 1 presents a diagrammatic summary of the inter-related frameworks. Hence, initial data analysis conducted via COM-B was further enriched by the use of the TDF.

### 2.2. Participants and Recruitment

Eligible participants were recruited via purposive and snowball sampling. Individuals that fulfilled the following criteria: (1) Parent to a teenager aged 13–18 years old; (2) of Middle Eastern background; (3) an Australian resident; and (4) able to provide informed consent, were invited to participate in the study. Recruitment flyers were shared on social media platforms such as Facebook, Instagram, and WhatsApp, making it available to existing public groups, friends, and connections within the networks of the researchers. Recruitment flyers were also posted on public noticeboards in the local government area and were distributed in areas where parents of Middle Eastern backgrounds congregate, e.g., local community and recreation centres. Passive snowball techniques were exclusively used, whereby eligible participants who had signed up informed others about the study. Participants who were interested were able to scan a QR code that was included on the flyer that redirected them to a screening questionnaire on the Research Electronic Data Capture (REDCap, Vanderbilt University, Nashville, TN, USA, 2004). If eligible, they continued on to complete a short demographic questionnaire that collected information on the number of children aged 13–18; the age of their children; the parent’s age, postcode, email, and gender; and if the participant was a single parent. The socioeconomic status of the participants was determined by cross-referencing the postcodes of the participants against the Socio-Economic Indexes for Areas (SEIFA) as listed by the Australian Bureau of Statistics (ABS) [30]. Upon completion, the participants were sent an automatic email with the participation information statement and consent form attached. All participants provided consent and were contacted by one researcher (NH) who organized the dates and times for the interviews to be conducted via telephone or Zoom, as chosen by the interviewee. The interviews were conducted from the 16 November 2020 to 28 July 2021.

### 2.3. Procedure and Data Collection

The semi-structured interviews, which ran for about 60 min, were conducted by the lead researcher (NH), a female accredited practicing dietitian and PhD candidate who is of Middle Eastern background. The interviews used a guide of 35 questions for physical activity and screen time (100 questions on nutrition and obesity have previously been reported) [23], which were developed a priori (Table 2). The questions were determined through the agreement of two researchers (NH and MAF) and were structured based on the COM-B [21] model and the TDF [22,27,28,31]. Sample size was not defined a priori; rather, we aimed to conduct sufficient interviews to obtain a rich data set based on the concept of information power [32].

Time was allocated at the beginning of the interview to allow for a brief introduction and rapport building. Participants were reassured that their personal data would be kept secure and de-identified, and final consent was obtained before the recording of the session. Questions were asked regarding enablers and barriers on physical activity and screen time practices. Upon the completion of the interview, the participants were reimbursed with an $20 (AUD) voucher for their time.

### 2.4. Data Analysis

All sessions were digitally audio-recorded and transcribed verbatim by one researcher (NH). A two-step transcription process was employed to minimise errors, with a second hearing to correct any errors if present. Transcripts were not returned to the study participants. Data such as the transcriptions and audio data files were stored on the Research Data Store (RDS, University of Sydney, Sydney, Australia, AU, 2020/21). De-identified transcripts were imported into the NVivo Pro (12th Edition, QSR International, Doncaster, Australia, 2018) qualitative data analysis software for thematic analysis. A reflexive thematic analysis was used to construct the coding framework, which consisted of a combination of deductive (themes arising from the interview questions) and inductive approaches (themes emerging from the data during analysis). Data were analysed through the following five steps, which were in line with the framework method, where (1) the transcripts were examined independently (NH, YS); (2) initial themes were constructed independently using the COM-B subcomponents and 14 TDF domains (NH, YS); (3) the themes and organizing data under each relative theme and sub-theme were coded (NH, YS); (4) the themes were reviewed and refined (NH, YS, MAF); and (5) the data were summarised and reviewed using representative quotes (NH).

## 3. Results

Of the 58 participants who were interested in completing the screener questionnaire, 57 were eligible. Of the 57, 26 agreed to be interviewed. Reasons for not participating included inability to reach a mutual time and a non-response from those interested post-sign up. It should be noted that interviews were conducted at times during government lockdowns because of the COVID-19 pandemic, which could have contributed to the non-response rates. Participants were recruited from November 2020 to July 2021. The lead researcher continued to recruit participants until it was determined that no new topics were emerging from the interviews being conducted and an information-rich data set was obtained. The duration of the interviews ranged from 40 min to 1 h and 33 min and covered nutrition behaviours, as reported elsewhere [33], as well as the questions regarding physical activity and sedentary, which comprised 30–40% of the interview. All of interviews were conducted in English, except for one parent, who was more comfortable speaking in a combination of Arabic and English. All of the parents were female, and the majority of mothers were from low to middle SES areas (n = 19), as determined by the SEIFA, aged 40–49 years of age (n = 17), not a single parent (n = 21), and the parent to one or two adolescents (n = 20) aged 13–14 years (n = 14). During the interviews, it was found that a majority of mothers were of the Islamic faith and were of a Lebanese background although no information was specifically asked during the interviews.

The findings are summarised below under the six domains of the COM-B model and subthemes from the TDF with representative quotes presented in Table 3. No part of the interview was coded to Psychological Capability for Memory, Attention, and Decision Processes and Reflective Motivation for Intentions and Beliefs about Capabilities.

### 3.1. Psychological Capability

#### 3.1.1. Knowledge

Physical activity: Parents expressed a high level of awareness of the need for the adoption of healthy physical activity behaviours in adolescents, irrespective of absence or presence of weight concerns. However, parents were concerned over lack of physical activity (PA), especially in girls and later age adolescents. Physical movement in adolescents was largely attributed to extracurricular organised sports such as Oz tag, basketball, swimming, and soccer 1–2 times per week. Such activities usually demanded extra time for the maintenance of physical endurance and strength, which was completed via gym and training sessions 2–3 times a week, which was mostly reported in boys. Parents were eager for participation, claiming at least half an hour to one hour per day should be dedicated to PA. One parent relayed that 15 h per week of PA by their adolescents would be substantial to meet physical activity guidelines.

Screen time: The mothers demonstrated great apprehension over excessive screen time and were aware of the link between excessive screen time and weight gain because of lack of physical activity. Whilst excessive screen time was linked to several devices such as TVs and iPads, smartphones were the most ubiquitously used and the most frequent source of extended periods of leisure screen time. The mothers claimed that adolescents were addicted, with at least 2–3 h a day spent consumed with their smartphones. Many reported several hours a day, reaching 8 h or more during holidays, a number that was exacerbated by the COVID-19 pandemic. In addition to health concerns, the parents had intimate knowledge of the dangers of social media, a major source of worry. The parents affirmed that if they were able to exact influence on leisure screen time, they would limit it to 1 h. For the majority of parents, this remained elusive in the context of the digital revolution and the inherent and prevalent use of screen devices.

#### 3.1.2. Behavioural Regulation

Physical Activity: The parents aimed to increase physical activity by introducing at least half an hour per day of activities after school and signing their children up for extracurricular sporting activities; however, this was more successful in boys. Girls who did listen and take part in sports usually attended the gym with theirs parents and friends and joined boxing classes. The parents also tried to take their children to the park when they were free to encourage at the very least walking.

Screen Time: The parents demonstrated multiple regulatory strategies to limit screen time and content. This included only allowing phones once they reached 14/15 years of age, enabling child-friendly settings on devices, banning social media, or limited access using the mother’s phone and social media accounts. Some parents did not allow the devices to be taken to school, and the majority set a schedule during the week to minimise screen time. They also ensured that they had passwords to devices and made sure to monitor their use, encouraging their children to use devices in open rooms such as the living room, especially for girls. Some also had curfews, where the adolescents handed in their devices before retiring for the night.

#### 3.1.3. Cognitive Skills

The parents displayed various skills in helping to mitigate excessive screen time and content, including child-friendly settings and set times, but expressed the need for parental support in maintaining these rules, especially for older adolescents who had more independence and who were more likely to carry a smartphone. The COVID-19 pandemic as well as the abilities of a smartphone to be used for watching, gaming, and communicating with their friends rendered any regulatory efforts fruitless.

### 3.2. Physical Capability

#### Physical Skills

The parents reported that their children displayed the necessary skills to engage in all sorts of physical activity whether it be gym sessions or organised sporting activities. They did however express concerns over a decline in physical skills due to excessive screen time behaviours. Most parents had to initially encourage their children to partake in PA opportunities.

### 3.3. Reflective Motivation

#### 3.3.1. Beliefs about Consequences

The parents unanimously agreed that excessive screen time was linked to a myriad of health consequences both in the long and short term. Excessive screen time was attributed to a lowering of IQ, concentration issues, headaches, neck and back problems, laziness as well as dissociation from family. The parents were also concerned with the effects of radiation absorbed after hours of exposure at an age where the brain is still developing. Disruptions to emotional regulation were also cited, linking anger management problems, agitation, and emotional withdrawals to excessive screen time and gaming. Similarly, the mothers identified laziness, excessive weight gain, agitation, and mood disruptions such as depression to a lack of physical activity. Loss of social skills was linked to both excessive screen time and lack of physical activity. Lastly, the parents were cognizant of the social dangers that are mediated through excessive screen time via social media, with unprecedented exposure to a wide variety of influences that could affect adolescent mental, physical, and social health.

#### 3.3.2. Social/Professional Role and Identity

Without exception, the parents believed that they had a seminal role in encouraging physical activity and moderating screen time. It was also reported to be easier to encourage PA if the fathers were also active and shared a responsibility in raising an active family. However, they believed such efforts needed to start from an earlier age to ensure a smooth transition when they reached adolescence, which many parents described to be a very difficult process. They also stated that they needed greater support in the face of the COVID-19 pandemic and its long-term implications, which inherently restricted activity and elicited greater opportunities for screen time. The parents acknowledged current government initiatives such as AUD 100 vouchers to encourage sport participation but believed they could do more to ensure more consistent physical activity, considering the expenses of sport clubs such as membership, attire, etc. The parents also highlighted that such vouchers were limited for use with specific associations, whereas they would like to use the voucher to encourage other forms of PA such as purchasing a bike to be ridden in the local area. They underlined the need for schools and local communities to work together and stated that current sports initiatives at schools were basic and only included sports once per week, if any, especially in Islamic schools.

#### 3.3.3. Goals

All of the parents maintained that healthy lifestyle behaviours such as engaging in physical activity and reducing screen time was of the utmost priority for a multitude of mental, physical, and social reasons. However, they suggested earlier intervention to be vital to allow older adolescents to foster these habits to be implemented out of their own desire, autonomy, and sense of responsibility towards themselves.

#### 3.3.4. Optimism

All of the parents affirmed their confidence in the need for a shift from current physical inactivity and excessive screen time and believed any changes, even if they were small, would be welcomed and would elicit a myriad of health benefits, especially tracking into adulthood. They believed any change that was representative of the inherently physical activity lifestyle that the parents themselves enjoyed was vital but were much more optimistic that they could be encouraged and maintained in the context of the current physical and social environment as well as in the COVID-19 pandemic without community, government, and school support.

### 3.4. Automatic Motivation

#### 3.4.1. Emotion

The parents relayed their deep fears in attempting to raise their children in a society and environment that promoted laziness, excessive screen time, and lack of movement. Particularly, the mothers were distressed about the effects of the ongoing lockdowns during the pandemic, which isolated adolescents, forcing them to resort to screen devices for connection and kept them indoors. The relationship with screen devices was characterised as “addiction” and was regarded with great disdain. They wished their children could be an inherently physical active generation as they were. The parents identified the long-term consequences of the pandemic, with some children becoming teenagers during this time, a defining period, and were afraid that such excessive screen time and a lack of PA would be deemed to be normal. The parents revealed that their exasperation in failing to limit excessive screen time and encouraging physical activity. They reported having many conflicts with their children, especially over screen time, with time restrictions being placed on devices often leading to severe tantrums and withdrawal symptoms that were similar to those of other “addictions”. The parents were highly confident that any reduction in screen time and an increase in physical activity would yield many health benefits. However, they were equally despondent on the current state of adoption of these lifestyle behaviours and pessimistic if change could be implemented, especially for screen time behaviours.

#### 3.4.2. Reinforcement

The parents attempted to enforce a routine with children regulating PA and ST to set times after arrival from school on weekdays. Naturally, parents allowed greater access to screen devices and entertainment services on the weekend. The parents did not apply strict rewards or punishments, but many resorted to extreme measures such as turning off the power supply, Wi-Fi, and confiscating screen devices if the children were deemed to cross limits and enjoy excessive screen time. The parents also confiscated devices as a means of punishment for behaviours that were unrelated to excessive screen use, such as poor academic performance and ethical issues. They did not reward or punish physical activity behaviours.

### 3.5. Social Opportunity

#### Social Influences

The parents emphasised the influence of macroscopic factors such as changes in societal values and the “digital revolution”, which forever revolutionised the communications and physical activity landscape. The parents claimed that their era inherently encouraged physical activity due to the safer use of public transport and a lack of screen devices to distract from normal play. There was no need for sports clubs, and incidental activity was more than enough to at least satisfy current physical activity guidelines. The mothers expressed their dismay on the shift of societal values/moral fabric of society and the deterioration of trusted shared spaces in village-like communities, which were primarily responsible for their childhood physical and sport engagement, which is in stark contrast to the current safety concerns in local neighbourhoods. As opposed to a collectivist culture that is representative of the culture practiced amongst their first generation parents and abroad, today’s increasingly individualistic culture in conjunction with safety concerns means that parents are much less likely to allow their children to join local activities unchaperoned.

The mothers proclaimed that friends had a great deal of influence on leisure screen time behaviours via the use of electronic devices to access social media and other entertainment platforms such as YouTube. The mothers were more likely to associate peer pressure in girls, resulting in them participating in telephone calls and Instagram, whereas boys were more likely to partake in shared gaming and entertainment activities on PlayStation^®^ and Xbox^®^. Influence from friends to join physical activity initiatives was more likely to occur amongst boys, such as gym training for physical stamina for organised sports as well as for enhancing physical appearance. Parents reported a negligible influence from friends on girls.

### 3.6. Physical Opportunity

#### Environmental Context and Resources

The parents reported the presence of government-funded healthy lifestyle programs such as Go4Fun for children aged 7–13 years who were suffering from obesity [34] but complained that such programs were not weight-inclusive or culturally responsive. Furthermore, they stressed the need for such programs to continue throughout high school to target adolescents, citing a dearth of any physical engagement with students and parents outside the mandatory Personal Development, Health, and Physical Education (PDHPE) classes. Some parents did mention after school sports programs run by external organisations but reported high a susceptibility to drop out given school changes and high commitment demands, especially for mothers of more than one adolescent. The mothers defined current Islamic school engagement with sports and physical activity to be rather “tokenistic” and called for the sincerer and more serious incorporation of movement instead of one-off events, especially for girls and for later children in their adolescent years. The parents communicated their concerns for excessive screen time given the shift to online education in light of the COVID-19 pandemic. Many parents described plentiful nearby parks and recreational facilities to encourage engagement in physical activity but also declared safety concerns, especially those hailing from low SES neighbourhoods. Parents only allowed chaperoned visits to the park, which were sometimes limited due to busy schedules. Many attempted to engage their children in organised sports and to become members of sport clubs; however, similarly, the parents reported that this was less likely to be achieved with girls and that the boys that did join eventually left due to lack of interest with older age.

The parents claimed that the physical layout of the home, regardless of size, was not a deterrent for physical activity due to the provision of local parks and facilities. However, the presence of several electronic devices in the household was reported to be linked to a continuous and addictive use of social media. Consequently, the parents resorted to ground rules, as detailed above.

## 4. Discussion

Parents struggle to grapple with the current digitised age and the unprecedented forces that have been exacerbated by the COVID-19 epidemic that limit their ability to mitigate excessive screen time and to encourage physical activity. The parents also expressed deep fears that the current sedentary behaviours that were adopted during COVID-19 would become the norm and worried about long-term consequences for this generation of youth. They reported that physical activity was compromised and a fear for the safety of children travelling alone to neighbourhood parks. Differences by gender were apparent, with older girls being reported to be the least interested in school sports.

One key finding of this study is that despite proximity to parks and other sports facilities, the parents demonstrated extreme reluctance to allow park visitation in their absence. It must be noted that this finding is not unique to ME parents, and while most of the previous literature has concerned children younger than 13 years of age, these concerns are not being voiced by the parents of adolescents as well [35,36,37,38,39,40,41,42,43]. Reluctance to allow unaccompanied park visits has been reported to be more pronounced in socioeconomically disadvantaged areas. Children who can visit public places unaccompanied do have more park visits [44]. Park and playground features have been shown to be linked to more physical activity [45]. Social concerns pertaining to the physical environment were raised regarding neighbourhood safety, fear of crime, and/or stranger danger were also established [46]. A cross-sectional study in the US revealed that the perceived neighbourhood crime rate was linked to children’s sedentary behaviour, but this was only found amongst parents from ethnic minorities [47]. Furthermore, a state-wide survey of Australian parents revealed that the perception of a child’s competence to travel safely was associated with parental fear, and having a female child and speaking a language other than English was linked to fear of strangers [48]. Parents participating in the Neighbourhood for Active Kids study indicated the need for safer traffic environments to mitigate fears and to further their children’s independent mobility [49]. Gender differences have been identified in the US [50,51]. It is important to promote park use by considering evidence showing positive associations between the availability of local parks and playgrounds and youth physical activity, but strategies to enhance safety should be mutually negotiated with communities.

Whilst evidence has shown the need for interventions to target such parental behaviour, the promise for combined school and community interventions [52] hold their appeal to placate fears by utilising school and community physical activity spaces in the presence of trusted community and school personnel. The need for interventions based in the community rather than only the school setting is highlighted given that very few exist in prevention intervention research [8]. The association between BMI and locked schools has also indicated that efforts to halt obesity should include increasing weekend schoolyard accessibility to allow adolescents to play sports [40].

A recent systematic review assessing built environment in programs to promote PA in Latino children and youth highlighted the need to capitalise on social ties and to engage youth in co-creating healthy environments using comprehensive methodological approaches that can address and establish context-specific priorities for each neighbourhood’s built environment and promotion of intramural activities [53]. Social Network Analysis (SNA) allows community social networks and norms to be integrated into PA interventions and has been used to assess potential mechanisms to achieve behaviour change. One study demonstrated that different targets in PA interventions (student opinion leaders’ vs. sedentary children) can achieve different goals (shift entire distribution of PA vs. increase PA levels in those most affected) [54]. Another purported framework is the Our Voice Model, which allows the engagement of stakeholders, families, and youth as citizen scientists to evaluate local social and environmental factors influencing PA participation in “real-time” using an application [55]. This has been found to strengthen community efficacy and the ownership of changing behaviours and their local environments and advocacy for a change in policies [56]. Such design would allow further comprehensive mapping and the consideration of factors such as street connectivity, which has been linked to PA [57]. One such project is being undertaken with adolescents from low-income neighbourhoods from multi-ethnic backgrounds in Sweden. The framework allowed adolescents to identify local features that promoted or prevented physical activity, the findings of which are being used to appeal to local decision and policy makers to improve local neighbourhoods to promote PA according to their specific needs [58,59].

There is emerging evidence that a school-based intervention integrated with health promotion activities yields positive results across green space use and competence, dietary habits, and resistance to substance use [60]. Green space interventions have shown potential to increase physical activity levels [61] and to lower BMI and prevalence of overweight [62], which makes it a possible effective design to be targeted in the future, given the nature of lifestyle behaviours such as diet and physical activity to cluster together. Indirect parental involvement is needed in the design of such interventions, as this study has shown that they exact influence over an adolescent’s opportunity for PA, as displayed in another study [63].

This is of particular importance given the documented link between a lack of physical activity and an increase in sedentary time because of increased screen time and emerging obesity. Focusing on community and sports initiatives in conjunction with advocating for specific policies targeting lower socioeconomic areas given higher fears of crime which limit children’s independent mobility is vital and will not only yield improvement in PA levels but will decrease sedentary time. There is evidence demonstrating that crime, a lack of quality sporting facilities, or neighbourhood options for physical activity may contribute to greater TV viewing among youth [64,65]. It was also shown that a school environment that promoted health policies, upheld mobile phone bans, and provided more time for physical education classes was associated with lower sedentary time and higher PA levels [66].

Excessive screen time was identified to be a significant precursor for parental frustration and worry, given its prevalent use in an inherently screen-dominated society. Its implications for health and its influence on increased sedentary behaviour and decreased physical activity was concerning. Whilst some parents were able to maintain routines and rules that guided screen use, the majority were unable to exact influence, and unlike for other behaviours, conflicts regarding screen time were more likely to be confrontational. This was demonstrated in a previous qualitative study which showed that parents felt able to limit adolescent access to sugary drink consumption but felt unable to yield control over adolescent screen time [67]. This is not surprising given its ubiquitous use, with 97% of Australian households with children aged less than 15 years having access to internet and with 91% of connected households sporting desktop or laptop computers, mobiles, and smart phones. Adolescents aged 15 to 17 years old were the highest proportion of internet users (98%) in Australia [68]. Parents complained COVID-19 heightened screen time, something that has also been reported by other authors [69,70]. The absence of screen rules amongst adolescents has been linked to overweight and obesity [20].

Several parallels were identified between the findings of this study and those of focus groups held with adolescents in a previous exploratory study by our research group [32]. In the current study, the parents revealed that boys were much more likely to be involved in sport and that girls were more likely to grow disinterested as they become older. The parents also communicated the lack of serious opportunities for continuous engagement in sports in schools, especially in Islamic schools, in particular for girls and older adolescents [71]. This was corroborated by the frustration that was expressed by adolescent girls regarding a lack of sporting initiatives and the experience of being denied access to sports facilities. Both attributed this experience to the “tokenistic” sports engagement Islamic schools that has been demonstrated [32]. The parents explained that schools place a greater focus on academia and neglect extracurricular activities. This interaction between culture and gender was also seen in immigrant families in a qualitative study that explored the influence of family values and traditions on adolescent physical activity in Qatar [72]. Acculturation and the influence of culture has been identified as having an influence on physical activity levels in Arab immigrant populations, but this has been largely restricted to adult audiences [73,74] and cannot be applied to adolescents who are usually second and third generation citizens of the host country.

To circumvent a lack of school opportunities, most parents enlisted their children into sporting clubs, which has been shown to contribute significantly to leisure PA [75]. However, such options only further widen socioeconomic disparities with barriers such as cost, accessibility, and lack of local facilities affecting families from the most disadvantaged backgrounds [76,77], something that was also identified by the parents in this research. A recent analysis has also shown that SES is inversely linked with sports expenditure, with an AUD 311 difference in median expenditure between the most and least disadvantaged populations [78]. However, the implementation of the Active Kids Voucher program in NSW is a clear demonstration of the power of upstream policies and programs to support physical activity participation. The provision of an AUD 100 voucher to cover the cost of structured physical activity registration was shown to increase the number of days physical activity guidelines were achieved, showing an increase from 4.0 days per week to 4.9 days per week after 6 months [79]. Moreover, it contributed to 42.4% of the total PA time per week and made a greater contribution to adolescents who were 15–18 years of age, spoke a language other than English, lived in lower SES areas, or who were obese [79]; however, it was shown that these programs were less likely to reach them [80,81]. The program is currently undergoing further evaluation [82], but the results to date suggest that these groups demand additional interventions to ensure that equitable benefits are reaped from such initiatives [83].

### Study Strengths and Limitations

This study possesses several strengths, including the rich and detailed data that were obtained using semi-structured interviews and the individual completion of the coding process by two researchers, which was then cross checked and overseen with a third in order to discuss and agree on themes. Among the limitations of the present study is the inability to capture paternal influence, as all of the interviews were conducted with mothers even though most of the families comprised two parents. Fathers also play a pivotal role in shaping adolescent physical activity behaviours [84]. Lastly, a majority of the mothers were from Lebanese backgrounds, and opinions of ME parents from other backgrounds may differ.

## 5. Conclusions

The findings of this demonstrate the need for intersectoral efforts across public health specialists, community organisations, and schools to deliver targeted real-world interventions that carry contextualised and culture-tailored messages. The interviews revealed that the majority of our participants were from the Islamic faith and of a Lebanese background. The need for interventions to be community-driven and inclusive of other subgroups within the ME community were recognized. Such efforts need to be complimented with upstream policies and public sector financial incentives to further reduce socio economic gaps in PA participation. The findings of this research reveal that Islamic schools may need to incorporate long-term sports programs, especially for girls. Further initiatives should centre on localised youth-driven approaches that aim to assess the local environment to inform context- and neighbourhood-specific interventions. Gender-specific and age-appropriate interventions should be prioritised to increase opportunities for girls and older adolescents, given their lack of PA worldwide [84].

## Figures and Tables

**Table 1 ijerph-18-12787-t001:** Thematic framework informed by the Capability, Opportunity, and Motivation Model of Behaviour and the Theoretical Domains Framework.

Behaviour/COM-B Component	TDF Domains
Psychological Capability	Knowledge Cognitive and Interpersonal Skills Behavioural Regulation Memory, Attention, and Decision Processes
Physical Capability	Physical Skills
Physical Opportunity	Environmental Context and Resources
Social Opportunity	Social Influences
Reflective Motivation	Goals Intentions Optimism Beliefs about Consequences Social/Professional Role and Identity Beliefs about Capabilities
Automatic Motivation	Emotion Reinforcement

**Table 2 ijerph-18-12787-t002:** Questions used in interviews to gather information on the capabilities, opportunities, and motivations of parents on adolescent physical activity and screen time behaviors.

Study Topic	Questions
Target Behaviour: Physical Activity and Leisure Screen Time	What do you think about the amount of time your child spends on the screen?
	How about time spent on physical activity?
	Have you ever heard excessive screen time might be associated with weight gain because the children are sedentary and not physically active?
	Do you have any rules about screen use with your children?
	Do you have any rules about time they spend being physically active?
	Does your child participate in any sports outside of school?
	Does your child enjoy physical activity?
	Do you impose limits on screen time?
	Are there any times that you take away devices to reduce screen time?
	Do you encourage physical activity?
	Do you see you have an important role to play in moderating your child’s screen time?
	Do you believe that you have an important role in encouraging physical activity?
	If not, do you think that you should?
	What might be your plan in this area?
	If you were to rate from 1 to 10 on how important/relevant screen time is, what would you say?
	If you were to rate from 1 to 10 on how important physical activity is, what would you say?
	What do you think will happen with your child’s mental and physical health if they have excessive screen time?
	What do you think is a reasonable amount of screen time?
	What do you think will happen with your child’s mental and physical health if they do not have enough physical activity
	What do you think is a necessary amount of physical activity per week?
	Does your child’s screen time affect you emotionally?
	Does your child’s time participating in physical activity affect you emotionally?
	Are there any conflicts regarding screen time?
	How does this affect you and your child?
	Do you enforce any routines around screen time?
	Do you enforce any routines around physical activity?
	Do you give any punishments or rewards for following the screen time rules?
	How confident are you that having less screen time will have a benefit for your child
	Similarly, how confident are you that having more physical activity will benefit your child?
	How does the physical layout of your home enable you to regulate your child’s screen use?
	How does the layout of your home enable you to encourage your child’s physical activity?
	How important is screen usage amongst children and their peers?
	How difficult does it make it to regulate your child’s screen time use?
	How important is physical activity amongst your children and their peers?
	Does this affect your ability to encourage physical activity?

**Table 3 ijerph-18-12787-t003:** Representative quotes from parents by key theme.

**Psychological Capability: Knowledge**
“They spend too much time, trust me; they binge watch everything and they’re really shocking. Yes, the worst possible mark in the world is my kids when it comes to screen time.”—P19, (F), 45 y, 1 adolescent “24/7, nonstop from the second they wake up and to the second they go to sleep; even if they’re in the toilet, they’re on their phones. They don’t get off it.”—P14, (F), 43 y, 1 adolescent “I reckon if they can get like two hours a day. That’d be amazing, like two hours of physically being outside, walking, running, being in the fresh air, like filling their lungs with like literally like fresh air, socializing, like skateboarding, swimming, whatever, like two hours a day. Phenomenal.”—P14, (F), 43 y, 1 adolescent “When they’re sitting on the Xbox all day. There is no exercise; the only any exercise they do is their voicebox.” (laughter)—P8, (F), 46 y, 1 adolescent
**Psychological Capability: Behaviour Regulation**
“Definitely. They’re not allowed to have any social media. That is one; second of all, they’re not allowed to watch anything while they’re in their bedrooms. Have to be in the lounge room with me so I can see what they’re watching I’m very strict list of things like that. I don’t let any kids sit in the bedroom. You’re only allowed in there if you’re sleeping or getting dressed. Other than that, everything else, you’re out in the lounge room in the main areas with me. Even homework. They’ve got a desk in the lounge room. They sit down and do their homework on.”—P5, (F), 37 y, 1 adolescent “We don’t have rules in place. But with the onset of the school holidays, which was last week, my daughter, I said to her that it’s compulsory basically that the family get an hour physical activity every single day. So, that could be a bike ride. It could be a walk somewhere, getting out of the house to drive to a park, to take a walk. And we’ve been trying our best to implement that as much as we can. But that would be the plan inshaAllah for the rest of the school holidays.”—P7, (F), 42 y, 1 adolescent “Yes, … So basically, she goes on her walks with me every single day. We would do a little bit of these machines provided in the park. So, we use those machines, for example, in regard to doing cycles and whatever we need to do. But she does a daily activity of at least forty to an hour every single day.”—P4, (F), 32 y, >1 adolescent
**Psychological Capability: Cognitive Skills**
“I think everyone should be at least active for at least an hour a day. Do you get what I mean? But with the, you can’t really force them; that’s the problem to be out there. Yeah, it’s just sad. So, I don’t know. It’s hard.”—P19, (F), 45 y, 1 adolescent “Very important for me to moderate it. But how do you moderate it? That’s my question. Like it’s always the question on the back of my mind. You know what I mean?”—P19, (F), 45 y, 1 adolescent
**Physical Capability: Physical Skills**
“I’m trying to join her to the gym so she can go with me to encourage her to lose some weight to become healthier, that she’s not interested. I put her in other activities like soccer, so she can move around, and it’s good for her social wellbeing and like just to be comfy and maybe help her reduce weight. So, I have some other activities maybe that will help her”.—P10, (F), 43 y, >1 adolescent “So, tennis every Monday and Tuesday; every Thursday and Friday, they do martial arts, and on Saturdays, swimming”.—P16, (F), 38 y, 1 adolescent
**Reflective Motivation: Beliefs about Consequences**
“I think well, I think they’d start to get more depressed, more upset, angry. I know for myself, if I don’t exercise, I’m grumpy all the time. I’m eating anything. I’m making excuses not to exercise or, you know, and just feeling bad all day. Like, physically and mentally because you’re not doing any movement, there’s no movement. There’s no, there’s nothing there, like …. And you notice the change in the anger now that he’s actually more calmer that he’s go to the gym. He’s you know, the weights he does or whatever he does. He’s releasing his tension in that. So, which is better. Like, you see a complete mood changes as well so yeah.”—P8, (F), 46 y, 1 adolescent “… a lot of kids, teenagers who they spent a lot of time behind the games and playing on the games on the Internet. And now when they’re older, they can’t socialize; they can’t because people they can’t socialise because of that. Because of that, they feel they are antisocial.”—P20, (F), 41 y, >1 adolescent “Definitely do damage, especially with TikTok and who they follow and who they listen to and idea if they start believing in influencers and what they say now has a big impact, has a big impact on them.”—P17, (F), 43 y, >1 adolescent
**Reflective Motivation: Social/Professional Role and Identity**
“Active? Yes. I try hopefully to get her to do it about four times a week. So, she does boxing twice a week, and sometimes, we have training twice a week for soccer and then soccer is over, even if I take her out, enjoy the summer because summer soccer, it’s so they don’t have to train for that you just go to the games. But yes, I try to take her out when we go on our walks. And then I’ll say that’s part of your exercise for the week. Or if she takes the ball and that kicks that around in the park while I go around walking, I know that she’s getting some form of sport in there or exercise. Yep.”—P9, (F), 42 y, 1 adolescent “I mean, I think at large schools and communities should be really involved in that regard, and especially with a Middle Eastern family, as you were saying, the rate is very high. I think both of them should be hand in hand in regard to getting the kids to move around; whether they have a what do you call it, they have the kids voucher. The thing is, by using that, everything is really expensive these days, so you can only use it for one activity and for one term. And to want to say if you want to continue on, like trying to get the kids health and things like that, it comes at a high cost and some parents can’t really afford it, so it becomes an issue then. It’s like, okay, then if I can’t afford it, then how am I supposed to deal with this? And I can’t use them like on activities such as buying them a bike, you get what I’m saying or something for them to use within the home? No, it’s got to be something to use within the government or the associations they’ve got. So, if they extended those vouchers, the things that them in regard to buying a bike or buying a basketball ring or buying something to get the kids moving even at home, I think that would help a lot more. It will take them out of the house. So that’s what I would do, you know, I mean, that’s what I did because my kids were sucking up and they were sitting in front of the TV; I ended up buying even a second-hand bike. Don’t get me wrong, I had to buy a secondhand bike to get them moving or I go to a secondhand basketball ring as well to get them outside playing. I ended up coming out to play with them. So, they feel like, yeah, I look, if my mom is doing it with me, so why not? So they get at least half an hour to an hour of physical activities, but it comes back; everything works hand in hand parents, community, and the government; if they all are connected, they, I think, in regard to the habits that kids develop and things like that becomes a lot easier to deal with.”—P4, (F), 32 y, >1 adolescent “And that’s something my husband I do agree on is the physical activity because my husband is very sporty as well. So, he’s very active and my kids see that. So, that kind of encourages them to be very active. So he goes to the gym, he plays soccer, he does all this sort of stuff, and they see that. And I love the fact that they say that he’s very active. So that kind of encourages them as well.”—P16, (F), 38 y, 1 adolescent
**Reflective Motivation: Goals**
“No, I’ll go with ten. It’s very important, especially for this generation. As I said before, everything at the moment is connected through the media and social media. And that’s they using within the schools. So, if it’s used in school, it’s got to be used on a home. So, it takes away a lot…very important. Like I said, being physically active can deter a lot of health issues and just maintaining a good body weight and so helps with the growth as well and things like that. So, I think it’s very important. Yes.”—P4, (F), 38 y, >1 adolescent
**Reflective Motivation: Optimism**
“I’m very confident she can’t see that. I don’t think many, many teenagers can at this age. But I am confident that, you know, this is the best thing for her at this point.”—P7, (F), 42 y, 1 adolescent “It benefits their health. And it also benefits. It stops them from being on the phone. It’s something that’s always good for you because any exercise is good.”—P12, (F), 48 y, >1 adolescent “I’m very confident that it will be better for her; obviously, not sitting on a PlayStation or whatever and moving her body is much more better than just, you know, doing nothing. So, I’m very confident. Hopefully that’s just with me some though without school because obviously she’ll do the sports at school as well. So there is more sport during the week, but with me outside, actually, I’d say more than five hours a week.”—P9, (F), 42 y, 1 adolescent
**Automatic Motivation: Emotion**
“I can’t control it anymore; that’s how I feel. Like, today, I lost my cool, I told him I’m going to disconnect the internet and keep the house without internet. I don’t care, do what you want, it’s over, if I see you sitting and playing the game. He’s not sitting on the game today because I swore by God, but how about tomorrow? He’s going to play on it, and I can’t go about everyday swearing the same oath by God. Like, it would mean I have to continue to do that; it becomes not normal, but last time I disconnected it and kept it under my pillow with me at night, so he told me I’m not going to do anything. I’m not going to study. I told him don’t study, it’s your loss. I don’t care.”—P26, (F), 55 y, >1 adolescent “Well, if you take their devices away, it’s like you kill them; you’ve taken away their life. “How can you do that?” Yeah, it’s, I think they are really hooked and addicted to the to the technology, to the virtual world, to the fake world … They don’t like to be told to get off their devices, put it that way, unfortunately. That’s why I have to sometimes literally lock their devices up so that they can actually just go out, just go out for an hour, who cares? Just do whatever you want, even if it’s just sitting in the sun, you know what I mean? Getting some Vitamin D doesn’t really matter. But yeah, it’s, I think answering these questions now is harder than maybe if you asked me without being COVID, maybe things would be different. It’s just yeah, it’s difficult.”—P19, (F), 45 y, 1 adolescent “No, it actually is really upsetting on me. Like, when I come home from work and see them outside, I’m so happy. But when I if I drive in and I don’t see their bikes outside and I know they’ve been inside all day and I have to get them out, it, like, frustrates me because I know exactly what they’ve been doing all day…Yeah, no they have full withdrawals. Y has an emotional breakdown. And the other one, if you ask her to give her phone, she has a fit. She’ll, like, run out of the room and hide in her room when you ask for her phone. Yeah. The other one has withdrawals, starts shaking like he’s a bloody ex junkie. “I need my PlayStation”. No they do. They get very emotional when you take their, like, phones off them.”—P14, (F), 43 y, 1 adolescent “But I don’t think they realize what effect it has on them due to it’s become a become a norm. And because it’s, become a norm, it’s no longer an issue. Because there are other things that if the teenagers are on their screen 24/7 are looking at things, stuff like that, they don’t relate it back to that being the issue. They relate it to ever getting their head done in or not allowed to do something, or do you know what I mean? So, it becomes other things, become mental, it’s not the screen time; the screen time is an escape for them.”—P11, (F), 45 y, >1 adolescent
“No, really? To be honest, it’s. It is a benefit having their screen time, but kids these days, they don’t have the screen time, they get more agitated. Do you get what I mean? Like, if you’re not giving them their phone and you’re taking it off them, they’re just running amuck and just breaking everything. I, they do; you don’t understand. It’s like withdrawal symptoms. That’s how I see it. Like, phones these days, they’re pretty. I think they’ve been addicted to it.”—P12, (F), 48 y, >1 adolescent
**Automatic Motivation: Reinforcement**
“Yeah, I do. Yeah. When I, when I, like, click it, as they say, all the devices go to the locker. I’ve got a little locker that I put all the devices in, and I just put it in there, and just leave it there.”—P19, (F), 45 y, 1 adolescent “I just don’t like them being in front of a screen for too long, so sometimes, it could go to three hours and then. And then. And then, I’ll say to them, that’s enough, like, you need to get up now, you’ve got to find something to do. It’s, it can’t be twenty-four, seven on a TV. And actually, just yesterday, I unplugged the PlayStation because they were on it on the weekend all weekend. That’s all they did was play PlayStation. So, I’ve literally unplugged it, and I’ve hidden it now, and they’re not allowed to play PlayStation because I just think that’s too much. It’s too much to, way too much, screen time for them of an evening; they might put a movie on and watch a movie and then go to bed. So, yeah, I agree. I definitely agree. Too much screen time would obviously lead to obesity if they’re not going to be active and getting up or doing anything. So, I do put the time restrictions on”—P16, (F), 38 y, 1 adolescent
**Social Opportunity: Social Influences**
“But sometimes, you know, it’s a hassle because of the influence of the friend. Like, you have is your son, who’s 12, and he doesn’t have a phone, where his friend is, he has all the devices. And sometimes, you allow your kids to play certain games and then you give him all access to all these, you know, the games; they’re destructive, you know. This struggle, it is a struggle when it comes to games really; it’s not easy to put a schedule and stick with it.”—P22, (F), 50 y, >1 adolescent “For school, they use screen to talk in the class, but for friends and usually by phones, they chatting and sending messages, but talking like this, zoom, or for example, video call no; she used the screen of the telephone only for chatting about school and homework and sometimes about silly stuff after school.”—P23, (F), 48 y, 1 adolescent “Yes, because conversation starters on Snapchat; other than Snapchat, then they’ve got all the social media and Instagram, TikTok, all these applications that are coming through. Just to make sure that the kids are addicted to that screen time by watching other people being silly or stupid or doing some idiot things, so it does have an effect because this is the talk of the town, basically. What are you watching? What are you doing? Have you seen this? Have you seen that? Yep. So, it plays a massive role.”—P4, (F), 32 y, >1 adolescent “Yeah. And the fact is that she’s not on social media; some of her friends are, not all of them, I think, is another way that we’re protecting her; so, protecting her mental well-being I think because I know the negative effects of social media has on young minds, young, impressionable minds, and she knows these reasons. We’re quite frank with her; she’s quite mature for her age as well, we feel. And that’s kind of what informed our decision as well. So, we’re not coming at it from like “No, absolutely not. You’re not allowed”, let’s rationalize why you know we made this decision and why we’d advise you to keep away from that, at least until, you know, your HSC is over and until we feel we haven’t told her obviously this. But until we feel, you know, she’s matured enough and knows enough about the world and about herself before letting something toxic like that into her life because I feel that’s probably the biggest challenge we face with teenagers is keeping them offline because it’s a fear of missing out. You know everyone else. What what’s everyone else doing? Snapchat as well, I think, which is also popular amongst teenagers because they want to know what everyone’s doing at every given moment. And it’s just I believe it’s a big waste of their, of their time and their energy in their youth. I think as a youth, you know, they’ve got so much potential for so much good and so much personal growth. And I think I would prefer that these things are embedded more deeply in my daughter before she kind of you know has an eye into just how toxic social media can really be for this age group.”—P7, (F), 42 y, 1 adolescent “So, boys and girls are different. Whereas girls will be on it chatting, in boys will be on it doing other things they are either playing a game or they’re watching things on YouTube or both. Yes, I rarely hear my boys talking to their mates unless they’re playing, say, Fortnite through the microphone, but they’re not really texting and chatting their friends as opposed to my daughter is.”—P17, (F), 43 y, >1 adolescent
**Physical Opportunity: Environmental Context and Resources**
“Well, I just say in Arabic, I just say “bisatlu” (go dumb), but anyway, you know what that means, but they’re just going to be like zombies, you know what I mean? But, yeah, I don’t know. I don’t know. It’s really hard these days to answer these kind of questions, particularly like last year and this year especially it’s just. They’ve entered their teenage years with COVID, you get what I mean, so it’s really hard.”—P19, (F), 45 y, 1 adolescent “I would love them to grow up the way we grew up, the way we were always out and about. We never sat at home, we never, we hardly watch TV. We hardly only at night. But they, like you, have TV on day and night. You know, there’s nothing to do because life is getting boring, you know, and then there’s not much out there. Even if you go and you want to buy, for example, toys in the beginning, we used to find heaps of stuff. But now you find nothing. All you find is games and the new Xbox. New things like that new Nintendo. That’s what you find, you don’t find something that their child can use their brain for.”—P11, (F), 45 y, >1 adolescent “I think that, I think that. The areas that we live in, I don’t think there’s a really including a place probably, more emphasis on the schools, particularly the Muslim school. I don’t think there’s enough encouragement to be active and to be involved in sport. I really think that Muslim schools need to look at that more carefully because there’s such a focus on academics. I think that kind of gets left behind. It’s not really much of a concern other than the tokenistic athletic carnival every year. And it’s really not enough effort going into sport at all.”—P7, (F), 42y, 1 adolescent “I, I think schools is, you know it’s very important. I would love that. I would love schools to implement more activity in their, you know, in their timetable. Like currently, my daughter only done, only has a sport once a week, and it’s not enough, and she’s not getting it at home as well. So, I think school would help a lot if they could, you know, provide a time during the day for the kids to get active because this is a problem across the board with the new digital, you know, era that we live in. So yeah, I think if schools could take that on board and help out, that’ll be great. And just parents, we need to be more, I think we need to take up take more responsibility, you know, and, uhm, direct our kids maybe towards more exercise.”—P7, (F), 43 y, >1 adolescent

## Data Availability

Due to privacy reasons in ethics, we cannot make the data publicly available, but please contact author if further information is required.

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
