# Peer review of "Enabling Better Physical Activity and Screen Time Behaviours for Adolescents from Middle Eastern Backgrounds: Semi-Structured Interviews with Parents"

_ijerph, 2021, doi:10.3390/ijerph182312787_

Round 1

Reviewer 1 Report

Reviewer comments:

Enabling better physical activity and screen time behaviours for
adolescents from Middle Eastern backgrounds: semi-structured interviews
with parents.

General comments:

The topic is very important, and the population researched an important one in terms of numbers.

 However, there is confusion and lack of clarity as to exactly who the parent population is Three terms are used- namely ME (Middle East), Islamic and Arab.   The ME is a large region, and the populations are not homogeneous, nor are all Islamic and/or Arab (Israel for example). Given that the authors have highlighted the importance of parental influences on adolescents' behaviors, it is important to be more precise as to whether it is only Arabic-speaking (Arab) parents, or only Muslim, or in fact anyone from the Middle East, be it Egypt/Israel/Jordan/Iraq/Turkey etc. As the authors are positing that "culture-tailored messages" are needed, this issue needs to be clarified.

Though the English throughout is good, there are several errors, and these should be corrected. The manuscript should be proofread again.

Line 60- should read adolescents and not adolescent's

Line 61- parents are…. holders   and not as is written

Line 62- have shown

Line 73- which are used

Line 76 – which would enable

Line 76- barriers and not barrier  as 2 are mentioned)

Lines 80-81. Consistency needed. Exploratory/explorative- This is also in line 448. It should be exploratory

Line 89- complementary, and not as written complimentary

Line 107- backgrounds

Lines 161-162 - should be information-rich

Lines 165,198, 212 – the majority. There may be other cases where "the" is missing.

Introduction:

Comparisons are brought with the US and Australia, yet a statement is made that is very general. To support this statement, further examples could have been brought, such as perhaps from the UK. In the discussion, only one other country is mentioned (Qatar, ref 69).

Methodology:

Is postcode, based on SEIFA, considered a good enough indicator of SES for the purpose of this study?

Lines 89-98. There is a description of two systems for analyzing responses. Maybe a table showing the correlation between the two systems would save some text, and give a clearer picture.

Discussion:

Well presented.

Lines 386-387- unclear.

Conclusion:

This section is weak, and should include specific issues which arose. It is far too general, and is not "convincing": enough regarding the issues which were raised, and which should be addressed, in terms of proposed interventions, government policy etc. .

Author Response

Dear Reviewer,

Thank you for your comments. Please find below a point by point response to address your concerns.

Reviewer comments:

Enabling better physical activity and screen time behaviours for adolescents from Middle Eastern backgrounds: semi-structured interviews with parents.

General comments:

The topic is very important, and the population researched an important one in terms of numbers.

 However, there is confusion and lack of clarity as to exactly who the parent population is Three terms are used- namely ME (Middle East), Islamic and Arab.   The ME is a large region, and the populations are not homogeneous, nor are all Islamic and/or Arab (Israel for example). Given that the authors have highlighted the importance of parental influences on adolescents' behaviors, it is important to be more precise as to whether it is only Arabic-speaking (Arab) parents, or only Muslim, or in fact anyone from the Middle East, be it Egypt/Israel/Jordan/Iraq/Turkey etc. As the authors are positing that "culture-tailored messages" are needed, this issue needs to be clarified.

Thank you for this comment. The selection criteria listed that individuals were to be of Middle Eastern Background. No limitations were placed on country of origin or faith. We did not specifically seek information on these fields but they inherently arose in discussions by the participants.

Though the English throughout is good, there are several errors, and these should be corrected. The manuscript should be proofread again.

Line 60- should read adolescents and not adolescent's

Thank you for this comment. This has been amended.

Line 61- parents are…. holders   and not as is written

Thank you for this comment. This has been amended.

Line 62- have shown

Thank you for this comment. This has been amended.

Line 73- which are used

Thank you for this comment. This has been amended.

Line 76 – which would enable

Thank you for this comment. This has been amended.

Line 76- barriers and not barrier  as 2 are mentioned)

Thank you for this comment. This has been amended.

Lines 80-81. Consistency needed. Exploratory/explorative- This is also in line 448. It should be exploratory

Thank you for this comment. Both have been amended.

Line 89- complementary, and not as written complimentary

Thank you for this comment. This has been amended.

Line 107- backgrounds

Thank you for this comment. This has been amended.

Lines 161-162 - should be information-rich

Thank you for this comment. This has been amended.

Lines 165,198, 212 – the majority. There may be other cases where "the" is missing.

Thank you for this comment. All 3 have been amended.

Introduction:

Comparisons are brought with the US and Australia, yet a statement is made that is very general. To support this statement, further examples could have been brought, such as perhaps from the UK. In the discussion, only one other country is mentioned (Qatar, ref 69).

Thank you for this comment. I have included an extra example from the UK in the introduction.

Methodology:

Is postcode, based on SEIFA, considered a good enough indicator of SES for the purpose of this study?

Thank you for this comment. Yes, it is very typical for the socio-economic index of disadvantage and advantage derived from postcode to be used for these purposes.

Lines 89-98. There is a description of two systems for analyzing responses. Maybe a table showing the correlation between the two systems would save some text, and give a clearer picture.

Thank you for this comment. We have inserted a table under this paragraph to show the relationship between the 2 inter-related frameworks.

Discussion:

Well presented.

Lines 386-387- unclear.

Thank you for this comment. This has been amended.

Conclusion:

This section is weak, and should include specific issues which arose. It is far too general, and is not "convincing": enough regarding the issues which were raised, and which should be addressed, in terms of proposed interventions, government policy etc. 

Thank you for this comment. I have amended the conclusion to include specific recommendations for policy, practice and research.

Reviewer 2 Report

Reviewer's comments on the manuscript entitled “Enabling better physical activity and screen time behaviours for adolescents from Middle Eastern backgrounds: semi-structured interviews with Barents” (ijerph-1461221). The aim of the study was capture the opinions of Middle Eastern parents on barriers and enablers to sufficient physical activity and limiting screen time behaviors in adolescents.

  1. Authors must adapt the manuscript to editorial requirements, in accordance with the instructions for authors.
  2. Often there is a missing space in the sentences, especially when there is a quotation.
  3. Please adapt the method of writing your references to the editorial requirements, in accordance with the instructions for authors.
  4. 4The authors should consider the inclusion of Table 1 and Table 2 in the supplementary material. 

Author Response

Dear Reviewer,

Thank you for your comments. We have tried our best to follow editorial requirements and will work with the editorial team to finalise changes.

- Nematullah Hayba, Yumeng Shi and Margaret Allman-Farinelli

Reviewer 3 Report

The article concerns an important research problem and I assess it positively. However, I propose a few additions that will improve its quality. The research gap should be clearly identified in the Introduction. The objective scope of the research is wide, so the aim of the article should be supplemented with detailed research problems (questions). The Discussion should indicate the contribution of research to the science system. The theoretical conclusions and practical implications of the research should also be deepened.

Author Response

Dear Reviewer,

Thank you for this comment. We have added information to show the research gap. We have supplemented the research aim to further elucidate the objective scope of this study. We have added to both the discussion and conclusion to incorporate your reviewer comments.

- Nematullah Hayba, Yumeng Shi, Margaret Allman-Farinelli